# A New LoRa-like Transceiver Suited for LEO Satellite Communications [note 1]

**DOI:** 10.3390/s22051830

**Published:** 2022-02-25

**Authors:** Mohamed Amine Ben Temim, Guillaume Ferré, Romain Tajan

**Affiliations:** IMS, University of Bordeaux, Bordeaux INP, CNRS UMR 5218, F-33400 Talence, France; guillaume.ferre@ims-bordeaux.fr (G.F.); romain.tajan@ims-bordeaux.fr (R.T.)

**Keywords:** DCSS, Doppler effects, IoT, LoRa, LPWAN, low-Earth orbit satellite, synchronization

## Abstract

LoRa is based on the chirp spread spectrum (CSS) modulation, which has been developed for low power and long-range wireless Internet of Things (IoT) communications. The structure of LoRa signals makes their decoding performance extremely sensitive to synchronization errors. To alleviate this constraint, we propose a modification of the LoRa physical layer, which we refer to as differential CSS (DCSS), associated with an original synchronization algorithm. Based on this modification, we are able to demodulate the received signals without performing a complete frequency synchronization and by tolerating some timing synchronization errors. Hence, our receiver can handle ultra narrow band LoRa-like signals since it has no limitation on the maximum carrier frequency offset, as is actually the case in the deployed LoRa receivers. In addition, in the presence of the Doppler shift varying along the packet duration, DCSS shows better performance than CSS, which makes our proposed receiver a good candidate for communication with a low-Earth orbit (LEO) satellite.

## 1. Introduction

A low power wide area network (LPWAN) is one of the most rapidly growing areas of the communication industry, especially in the Internet of Things (IoT) field. Indeed, according to [1], the share of LPWA connections will grow from about 2.5% in 2018 to 14% by 2023. By combining low energy usage, high noise resilience, and long range transmission, LPWANs are promising networks used to bring connectivity that fits the IoT aforementioned requirements. Both industry and academics are already making significant strides toward a mass IoT solution deployment. Indeed, multiple technologies with different physical and MAC layer standards have been defined to address constrained connected object challenges [2]. An ideal example of devices that fall under this category are sensors, used within smart cities, remote sensing, traffic control, supply chains, and so on. LPWAN technologies are accessible to support both licensed and unlicensed spectrum. Examples of 3GPP cellular technologies in a licensed spectrum include long-term evolution for machine type commutation (LTE-M) and narrow band IoT (NB-IoT). In the meantime, Sigfox [3] and long range (LoRa) [4] have reinvented connectivity for ongoing IoT ecosystem growth in unlicensed frequency bands.

Even with the wide coverage of LPWANs and the huge number of internet network operators, a limited area of the planet is currently connectable to the Internet. Indeed, terrestrial networks only cover 15% of the Earth’s surface [5]. Base stations and gateways simply cannot be deployed across oceans, deserts, or mountain tops, and they are not cost-effective enough to be installed in remote and sparsely populated areas. Therefore, low earth orbit (LEO) satellites, developed in recent years, can provide reliable communication services for places where there are no terrestrial networks. Currently, several works attempt to implement IoT communications with geostationary (GEO) satellites, such the work in [6], using the NB-IoT technology, and especially with LEO satellites. Indeed, several industrial and academic works have recently proposed deploying popular LPWANs in LEO satellite communications, such as nano-satellites lunched by Eutelsat to serve Sigfox’s generic IoT applications [7]. However, such communications lead to an increase in the complexity of the synchronization process, typically caused by the Doppler effects related to the satellite movements. Satellite IoT entail higher interference levels compared to ground-based IoT, due to the fields of view of LEO satellites, which allow connecting a huge number of end-nodes using several technologies, especially when considering ISM unlicensed bands. This issue is aggravated by uncoordinated access to the radio channel of the majority of LPWANs. We dealt with this problem in several of our previous works, in the case of LoRa communications. For instance, in [8], we provided an approach to “deal” the same-technology destructive collisions in LoRa. This method allows one to considerably enhance the throughput of LoRa-based networks since it makes it possible to decode up to four LoRa-like signals in a destructive collision. However, even with the huge number of devices that could be connected to a LEO satellite, more sophisticated methods should be implemented. We quantified in [9] the impact of an ultra narrow band interference of LoRa-like communications in ISM bands. Given this massive connectivity, the use of ALOHA-based random access protocols, as in the case of ground-based LPWANs, is a challenging task. Indeed, in the IoT communication with LEO satellites, the satellite needs to serve many terminals at the same time; thus, the probability of a packet collision is very important in such scenarios. Hence, an optimization of ALOHA could be implemented to reduce the number of lost or re-transmitted packets in such a context. For instance, the authors in [10] propose combining ALOHA with time division multiple access (TDMA) in order to reduce the probability of packet collision. However, this approach needs further synchronization processes between the satellite and end-devices, which would increase their energy consumption. It should also mention the potential use of enhanced spread-spectrum ALOHA [11] and Asynchronous contention resolution diversity ALOHA [12] for LEO satellite IoT, in order to reduce the probability of packet loss.

In this paper, we focus only on synchronization issues when considering LEO satellite communications using LoRa-like signals.

When LoRa communication is considered, the transmitted symbols are estimated in the frequency domain using fast Fourier transform (FFT) processing [13,14]. As a consequence:1.The presence of carrier frequency offset (CFO) causes a shift of all the Fourier transform peaks of a sequence of symbols to the right or the left of the desired frequency peak locations.2.A sampling time offset (STO) causes the emergence of two main shifted peaks in the spectrum, which lead to inter-symbol interference (ISI).

Based on these, accurate time and frequency synchronization are mandatory to achieve the theoretical sensitivity claims when using LoRa modulation [15]. Synchronization errors are some of the most important issues in IoT networks, especially when considering LEO satellite communication in unlicensed bands, due the random access to the radio channel, the low-cost local oscillators of connected devices, and the Doppler effects. Hence, to perform LoRa-like communications with LEO satellites, a sophisticated synchronization algorithm should be deployed.

In the literature, LoRa has been extensively studied in many aspects. For instance, several works [16,17,18] provided detailed studies on the capacity of the latter technology to cope with the requirements of LPWAN ground-based communications, such as the long range, low energy consumption, and interference resilience. However, few papers have addressed the issue of synchronization, especially the Doppler effect, when considering LEO satellites communications. For instance, authors in [14,19], propose to estimate the time and frequency offsets using a system of two equations produced by the estimation of the up-chirp symbols of the preamble and the down-chirp symbols of the SFD. However, this system could be solved only if the CFO is lower than $\frac{B}{4}$ [20], where *B* is the bandwidth of the chirped signal. This maximum CFO estimable could be exceeded in the context of the LEO satellite, as we do with the practical values in Section 4. In addition, due to the sensitivity of chirp spread spectrum (CSS) signals to time and frequency synchronization errors, the CFO and the STO parameter estimations must be very precise and tracking algorithms have to be deployed as proposed by [14]. To overcome the latter constraint, we proposed in [21] a novel synchronization approach associated to the well-known technique referred to as differential CSS (DCSS), but we did not deal with Doppler time variation. Furthermore, in the case of a time varying CFO, authors in [22] propose an algorithm to estimate the Doppler variation using the LoRa preamble. However, this method has a high complexity and cannot maintain its robustness for long packets size. In fact, this estimation is not perfect and, thus, a remaining Doppler variation can shift the symbols along the frame, which would significantly impact the decoding process especially of the lowest data rates. Moreover, the CSS modulation has been modified in [23] as symmetric CSS (SCSS) to ensure higher robustness against destructive collision. This modification is performed to make CSS more suitable to LEO satellite communication since the probability of collision would increase given the huge surface covered by the latter satellites. However, in this study the influence of CFO is ignored. To that end, the same authors propose in [24] the asymmetry CSS (ACSS). This approach offers better performance compared to CSS and SCSS in the presence of interfering signals. Nevertheless, the latter two works did not deal with the Doppler shift variable in time, which is the case for LEO satellite communications.

In the industrial field, Semtech recently developed the specification of her new physical layer long range frequency hopping spread spectrum (LR-FHSS) [25] to increase the capacity of LoRaWAN in dense and congested deployments. It has also been designed to support extremely long-range and large-scale communication scenarios, with a focus on reaching gateway devices installed on LEO satellites.

In this paper, we propose a novel algorithm to deal with the time and frequency desynchronization that impact the decoding process of LoRa-like signals in the context of LEO satellite communication. This work can be seen as an improvement of [21], which did not deal with the Doppler time variation. This improvement has led us to propose a new receiver adapted to such conditions. Hence, the main contributions of our work are to:Modify the CSS modulation in order to enhance its robustness to time and frequency synchronization errors, especially when the latter are time varying. Subsequently, our approach would allow to deal with Doppler shifts with much faster variations in time than the existent LoRa-like receivers.Release the constraint of a maximum allowed CFO of $\frac{B}{4}$ caused by the classical synchronization algorithms in LoRa [14,19,20,26,27]. To address this, the time synchronization is implemented regardless for the CFO. Currently, the frequency mismatch of local oscillators (LOs) between the transmitter and the receiver in LoRa-based communications do not reach this value maximum allowed CFO. However, this mismatch of LOs, combined with significant Doppler shifts, in the context of LEO communications, could lead to a CFO that exceeds the quarter of the bandwidth. Hence, with our approach, we can propose reducing the bandwidth of the chirped signals, without worrying about the occurrence of a CFO that exceeds the latter constraint, which would provide a gain in sensitivity (the actual choices of bandwidth for LoRa-like signal are based on the local oscillator precision to satisfy the $\frac{B}{4}$ constraint) and increase the capacity of LoRa-based networks.

To achieve all of these features, we propose a receiver based on the DCSS technique, which consists of transmitting symbols obtained by an integration processing (i.e., in each symbol time the cumulative sum of the current symbol and the previous ones is transmitted). At the receiver side, a differentiation is performed to recover the original symbols. As proved in [21], the latter differential processing allows the DCSS receiver not to perform a complete CFO estimation and tolerates more important time synchronization errors than existing algorithms in the literature. In addition, thanks to the differential process associated with an interpolation of the peaks in the Fourier transform, our proposal can tolerate important Doppler time variation. However, an estimation of this variation is needed for some configurations. Finally, the performance of our receiver has been validated with simulations in LEO satellite conditions.

The remainder of the paper is organized as follows. In Section 2.1, we provide a brief overview of LoRa PHY layer. The impact of synchronization errors on the symbol detection is detailed in Section 2.2. Building upon these models, we present, in Section 3, the DCSS technique and the proposed synchronization algorithm in six main steps. Before concluding our work, simulation results are proposed and interpreted in Section 4.

## 2. Synchronization Issues in LoRa

### 2.1. LoRa Physical Layer Principle

The LoRa PHY layer is based on a CSS modulation, which relies on sine waves whose instantaneous frequencies evolve linearly with time over a specific bandwidth *B*. These specific waves are called chirps. A raw chirp frequency varies linearly from an initial frequency $f_{i}$ to a final frequency $f_{f}$ during the symbol time *T*, with $B=|f_{f}-f_{i}|$. When $f_{i}>f_{f}$, the chirp is considered a down-chirp, while, otherwise, it is considered an up-chirp. Initially, the binary information flow to transmit is divided into subsequences, each of length SF. The set of SF consecutive bits constitutes a symbol. The number of possible symbols is hence equal to $M=2^{SF}$. SF indicates the spreading factor and the relation between the bit rate $D_{b}$ and the symbol rate $D_{s}$ can be written as: $D_{s}=D_{b}/SF$.

To distinguish between the *M* different symbols of the constellation, *M* orthogonal chirps have to be defined so that each symbol exhibits a specific instantaneous phase trajectory. This chirp is obtained based on the raw chirp and using $\gamma_{p}=\frac{S_{p}}{B}$, which allows performing a cyclic shift. It should be noted that $S_{p}\in ⟦0,\;M-1⟧$ is an integer coded on SF bits that corresponds to the transmitted symbol at time [(p−1)T, pT). The different trajectories are obtained by performing modulo *T* operations of a raw chirp. The raw chirp defined for t∈[0, T) is given by:(1)$$f^{0}\left(t\right)=B\left(\frac{t}{T}-\frac{1}{2}\right)$$

Then, the modulated chirp instantaneous frequency, corresponding to the $k^{th}$ transmitted symbol $S_{p}$, can be defined as:(2)$$\forall t\in \left[0,\;\;T\right),\;\;f^{S_{p}}\left(t\right)=f^{0}\left(\left(t+\frac{S_{p}T}{M}\right)\;\mathrm{mod}\;\;T\right)$$

We denote $f_{p}\left(t\right)$ the *p*th transmitted chirp by LoRa-like node, uniformly distributed within the set $\left\{f^{0}\left(t\right)\;,\;f^{1}\left(t\right)\;,\;\ldots ,\;\;f^{M-1}\left(t\right)\right\}$. Each chirp $f_{p}\left(t\right)$ is assumed to be transmitted during the period $t\in \left[(p-1)T,\;\;pT\right)$, thereby, the complex envelope of a CSS signal s(t) is a succession of random chirps, such that:(3)$$s\left(t\right)=\sum_{p=1}^{N_{s}}e^{j\phi_{p}\left(t-\left(p-1\right)T\right)}{𝟙}_{\left[(p-1)T,\;pT\right)}\left(t\right)$$
where $N_{s}$ is the number of transmitted symbols and the chirp $f_{p}\left(t\right)$, corresponding to an instantaneous frequency, such that $f_{p}\left(t\right)=f^{S_{p}}\left(t\right)$, can be expressed as the derivative of the instantaneous phase $\phi_{p}\left(t\right)$:(4)$$f_{p}\left(t\right)=\frac{1}{2\pi }\frac{d\phi_{p}\left(t\right)}{dt}$$

Therefore, we obtain for $t\in \left[0,\;T-\gamma_{p}\right)$:(5)$$\phi_{p}\left(t\right)=2\pi M\left[{\left(\frac{t}{2T}\right)}^{2}+\left(\frac{S_{p}}{M}-\frac{1}{2}\right)\frac{t}{T}\right]$$
and for $t\in \left[T-\gamma_{p},\;T\right)$:(6)$$\phi_{p}\left(t\right)=2\pi M\left[{\left(\frac{t}{2T}\right)}^{2}+\;\left(\frac{S_{p}}{M}-\frac{3}{2}\right)\frac{t}{T}\right]\;$$

According to [13], the transmitted symbols are detected by multiplying every *T*-long sequence of the received signal by the conjugate of a reference signal $x_{ref}\left(t\right)=e^{j\phi_{p}\left(t\right)}$, with $S_{p}=0$ (i.e., an unmodulated chirp). Moreover, the received signal should be sampled at $T_{s}=\frac{1}{B}$ in the demodulation stage [14,19]. A discrete-time version of $x_{ref}\left(t\right)$ sampled at $T_{s}$ is given by: (7)$$x_{ref}\left(n\right)=e^{j2\pi \left(\frac{1}{2M}n^{2}-\frac{1}{2}n\right)},\;\;\;n\in ⟦0,\;M-1⟧$$

Then, considering a perfect communication link, to estimate the $p^{th}$ transmitted symbol, a *M*-point FFT, Y[k, p], is performed as follows:(8)$$Y\left[k,\;p\right]=\frac{1}{\sqrt{M}}\sum_{n=0}^{M-1}\underset{d_{p}\left(n\right)}{\underbrace{s_{p}\left(n\right)x_{ref}^{*}\left(n\right)}}e^{-j2\pi \frac{nk}{M}}$$
with $s_{p}\left(n\right)=s\left(nT_{s}+\left(p-1\right)T\right)$, $n\in ⟦0,\;M-1⟧$ is the complex envelope of the $p^{\mathrm{th}}$ transmitted chirp and k∈⟦0, M−1⟧. After some calculations, $d_{p}\left(n\right)$ can be expressed as:(9)$$d_{p}\left(n\right)=e^{j2\pi n\frac{S_{p}}{M}}$$

Finally, considering a non-coherent receiver, the symbol estimate ${\hat{S}}_{p}$ is obtained as:(10)$${\hat{S}}_{p}=\underset{k\in ⟦0,\;M-1⟧}{argmax}\left(|Y\left[k,\;p\right]|\right).$$

Let us now consider a real communication link, to evaluate the STO and CFO impact on the demodulated signals.

### 2.2. Analysis of Imperfect Synchronization on Symbol Estimation

In this section, our objective is to derive and analyze a closed-form expression of the signal used to estimate the symbol in the presence of a:Time varying Doppler frequency shift, $f_{d}\left(t\right)=c_{d}t+v_{d}$, with $c_{d}$ the Doppler rate (DR) and $v_{d}$ is the Doppler shift;Uniformly distributed STO $\Delta \tau \in [-\frac{T}{2},\;\frac{T}{2})$.

It should be noted that the CFO Δf is equal to $v_{d}+v_{o}$, with $v_{o}$ being the frequency mismatch of LOs between the transmitter and the receiver.

Based on the latter notations, the continuous-time baseband received signal is expressed as:(11)$$y\left(t\right)=\sqrt{P}s\left(t-\Delta \tau \right)e^{j(2\pi \Delta ft+\theta_{0})}e^{j2\pi \int_{0}^{t}f_{d}\left(\tau \right)d\tau }+w\left(t\right)$$
where s(t), t∈R, is the continuous-time version of s(n), *P* is the received signal power, $\theta_{0}$ is the initial phase, and w(t) is the complex additive white Gaussian noise (AWGN) signal with $\sigma_{w}^{2}$ its variance. It should be noted that we considered here a non-frequency selective channel, which makes sense when LPWANs are considered, and even more for LEO communications.

To correctly obtain the radio frequency signal, and due to the CFO and the Doppler shift, the analog to digital converter (ADC) output signal should be sampled with $f_{s}^{\prime }$ greater than the Nyquist rate $f_{s_{min}}=B$, with an oversampling $\alpha =f_{s}^{\prime }/f_{s_{min}}$. However, to be compliant with the low complexity of the CSS demodulation principle [13], we consider in the following the sub-sampled signal at $T_{s}=\frac{1}{f_{s_{min}}}$. Indeed, as it will be explained in Section 3, our proposed synchronization algorithm works at $T_{s}$ contrary to the solution proposed by [22]. The signal sampled at $f_{s}^{\prime }$ is just used to perform a precise time alignment of the signal when the fractional part of the STO is estimated. The discrete-time version of the received received signal can be expressed as:(12)$$y\left(n\right)=\sqrt{P}s\left(nT_{s}-\Delta \tau \right)e^{j2\pi (\Delta fnT_{s}+\frac{c_{d}{\left(nT_{s}\right)}^{2}}{2})}e^{j\theta_{0}}+w\left(n\right)$$

To perform our analysis, we propose to focus our attention on the decoding process of the *p*th transmitted chirp. We notice that, in the presence of a timing offset, the signal processed by the FFT at the receiver is composed of two consecutive chirps as illustrated in Figure 1.

Thus, the signal in the *p*th *T*-long sequence can be expressed, after the dechirping process, as follows:(13)$$\;\begin{array}{cc} z(n,\;p) & =y\left(n,\;p\right)x_{ref}^{*}\left(n\right)\\ \; & =\sqrt{P}\left(v_{p-1}\left(n\right)+v_{p}\left(n\right)\right)+w\left(n\right)\; \end{array}$$
where y(n, p) = y(n+(p−1)M) $\forall n\in ⟦0,\;M-1⟧$. If we define $L\;=\;\lfloor \frac{\Delta \tau }{T_{s}}\rfloor$ as the floor value of the discrete time offset, the two signal components of z(n, p) can be written as:A contribution of the ${\left(p-1\right)}^{th}$ transmitted chirp during the time interval ⟦0, L−1⟧;
(14)$$\begin{array}{cc} v_{p-1}\left(n\right) & =s_{p-1}\left(n+M-\frac{\Delta \tau }{T_{s}}\right)e^{j2\pi \left(\Delta f+\frac{c_{d}\left(nT_{s}\right)}{2}\right)nT_{s}}x_{ref}^{*}\left(n\right)\\ \; & =e^{j\theta_{1}}e^{j2\pi n\left(\frac{S_{p-1}-\frac{\Delta \tau }{T_{s}}+\left(\Delta f+\frac{c_{d}\left(nT_{s}\right)}{2}\right)T}{M}\right)} \end{array}$$A contribution of the $p^{\mathrm{th}}$ transmitted chirp during the time interval ⟦L, M−1⟧;
(15)$$\begin{array}{cc} v_{p}\left(n\right) & =s_{p}\left(n-\frac{\Delta \tau }{T_{s}}\right)e^{j2\pi \left(\Delta f+\frac{c_{d}\left(nT_{s}\right)}{2}\right)nT_{s}}x_{ref}^{*}\left(n\right)\\ \; & =e^{j\theta_{2}}e^{j2\pi n\left(\frac{S_{p}-\frac{\Delta \tau }{T_{s}}+\left(\Delta f+\frac{c_{d}\left(nT_{s}\right)}{2}\right)T}{M}\right)} \end{array}$$where $\theta_{1}$ and $\theta_{2}$ represent two constant arguments, which have an impact on the symbol estimation in a presence of time synchronization errors due to the phase discontinuity created in the signal processed by FFT. As shown in (Equation 13), when the timing alignment of the received signal is not performed, ISI occurs. In the following subsections, we analyze the impact of the CFO, the Doppler shift and the STO on the symbol estimation.

#### 2.2.1. Impact of the CFO and the STO on Symbol Estimation

When the received signal is affected by a CFO, the argmax of all the FFTs would be shifted by ΔfT=C+ν, with

C=⌊ΔfT⌉ is an integer offset that shifts all the symbols;ν∈[−0.5, 0.5) is the fractional part of the CFO that shifts the spectrum line between two frequency bins, effectively making a sinc kernel appears in the frequency domain.

In the presence of the STO, an ISI occurs as depicted in Figure 1, which leads to the emergence of two cardinal sines with positions shifted by $\frac{\Delta \tau }{T_{s}}=L+\lambda$, where λ∈[0, 1) being the fractional STO. In addition, λ may cause a phase discontinuity of the modulated chirps [14], which implies a biased FFT processing.

For more details on the impact of the CFO and STO, readers can refer to [21,27].

#### 2.2.2. Impact of the Doppler Rate on Symbol Estimation

When only the DR is present (i.e., {Δτ, Δf}=0), we observe an uncompensated frequency offset that varies linearly with time at a slope $c_{d}$. For the sake of simplicity and to qualitatively understand the effect of this DR, let us approximate this linear variation as constant over a symbol time and changing from symbol to symbol (i.e., $f_{d}\left(nT_{s}\right)=f_{d}\left(pMT_{s}\right),\;\forall n\in ⟦pM,\;(p+1)M-1⟧$). Under this assumption, the frequency corresponding to the maximum amplitude of the FFT, when performed on consecutive symbols, will increase or decrease linearly (depending on the sign of $c_{d}$). Hence, (Equation 13) can be written as:(16)$$\begin{array}{cc} z(n,\;p) & =\sqrt{P}e^{j2\pi n\left(\frac{S_{p}+\frac{f_{d}\left(pMT_{s}\right)}{2}T}{M}\right)}+w\left(n\right)\\ & =\sqrt{P}e^{j2\pi n\left(\frac{S_{p}+\frac{c_{d}p}{2}T^{2}}{M}\right)}+w\left(n\right). \end{array}$$

Then the *M*-point FFT of z(n, p) gives:(17)$$Y\left[k,\;p\right]=\sqrt{\frac{P}{M}}\Gamma_{M}\left(k,\;S_{p}+\frac{c_{d}p}{2}T^{2}\right)+W\left[k\right]$$

As depicted in (Equation 17), the argmax of the FFT is shifted from symbol to other. We notice that the signals with the highest symbol times are more sensitive to the effect of the Doppler rate.

### 2.3. Insights on Strategies Used to Synchronize LoRa Signals

In order to properly measure the contribution we make in this paper, we recall, in this section, the main principles of the synchronization methods commonly used in LoRa [14,19]. It must be recognized that the synchronization method developed in [14] is very clever and offers an excellent compromise between the performance and the implementation complexity. However, the low computational complexity of the synchronization proposed by the latter work leads to the constraint of the maximum CFO estimable of $\frac{B}{4}$ [20,26].

To understand the synchronization process of LoRa, which leads to the latter constraint of maximum CFO estimable, it is mandatory to give a brief overview on the structure of the specific LoRa preamble used in this purpose.

#### 2.3.1. Structure of the Synchronization Signal

The signal transmitted by LoRa node starts with a preamble composed of $N_{p}$ up-chirps, which are exploited to detect the presence of a LoRa packet and to perform the time and frequency synchronization. $N_{sw}=2$ special modulated symbols known as synchronization word “sync word” are used to verify the accurate synchronization of the received frames (it is used also as a network identifier). The synchronization sequence end by two and a quarter down-chirps, known as the SFD, which help the time and frequency synchronizations [19].

#### 2.3.2. Synchronization Process in LoRa

Given the specific structure of the synchronization signal as presented in the previous paragraph, an estimation of the integer parts of the STO and the CFO (*L* and *C*, respectively) can be jointly performed. As explained in [14,19,28], a system of two equations using the estimated argmax of each FFT module in the preamble and the SFD is used to that end. If we denote the latter estimated frequencies ${\hat{S}}_{up}$ and ${\hat{S}}_{down}$ respectively, we have:(18)$$\begin{array}{c} {\hat{S}}_{up}=\left(-L+C\right)\;\;\mathrm{mod}\;M\\ {\hat{S}}_{down}=\left(L+C\right)\;\;\mathrm{mod}\;M \end{array}$$

Combining the two equations of (Equation 18), *L* and *C* can be easily determined. However, since $\hat{C}=\frac{{\hat{S}}_{up}+{\hat{S}}_{down}}{2}$ and ${\hat{S}}_{up}+{\hat{S}}_{down}$ is also modulo *M*, $\hat{C}$ can only be defined modulo $\frac{M}{2}$. Given the definition of *C*, Δf should be modulo $\frac{B}{2}$. As a result, the receiver will be able to recover a CFO only in the range $\left[-\frac{B}{4},\;\frac{B}{4}\right]$. Nevertheless, the authors in [22] have proposed a new method of synchronization that allows to overcome the latter constraint. To this end, they proposed using the sampling frequency $2f_{min}$, so that the estimated FFTs argmax are modulo 2M. This processing would resolve the latter problem, but it is done at the expense of the computational complexity.

Although this limit of the maximum CFO estimable is not really a problem, with regard to the deployed bandwidths and the carrier frequencies and the local oscillator precision, it prevents reducing the bandwidth of the transmitted signals. In fact, if the bandwidth is reduced, it is more likely to obtain a CFO that exceeds $\frac{B}{4}$, especially when considering LEO communications. The eventual bandwidth reduction makes it possible to increase the sensitivity of the receiver, as we will see later in this paper. It also leads toward increasing the number of possible channels, which could increase the capacity of the technology. However, it reduces the data rate and, thus, increases the time on air of the packets, which increases the probability of collision. Some applications, such as satellite IoT, lend themselves well to this need for long range communications. The contribution developed in the following section is not limited to this application case, but is particularly well adapted to it.

## 3. Proposed Transceiver

In this section, we detail the well-known differential process that is applied to the CSS modulation. Moreover, since our receiver has to deal with the time varying Doppler shift, we propose an additional processing to more precisely estimate the argmax of each FFT module. Finally, we detail all of the steps implemented by our synchronization algorithm.

### 3.1. Differential Chirp Spread Spectrum

#### 3.1.1. Principle

Based on the previous analysis, we propose an enhancement of the LoRa symbol generation process and we then show how it makes the detection of the received symbols robust to synchronization errors. Our idea, inspired by the principle of differential digital modulation techniques, consists of transmitting the value of the symbols, not directly, but rather their cumulative sum, so that, at the receiver, they can be retrieved by differentiation. In the following, we call this method of digital modulation the differential chirp spread spectrum (DCSS). Based on this, the DCSS transmitter consists of sequentially generating chirps based on the symbols $D_{p}$ obtained as follows: (19)$$D_{p}=\left(S_{p}+D_{p-1}\right)\;\mathrm{mod}\;M\;\mathrm{for}\;p\geq 0,$$
where $S_{p}$ has been defined as the LoRa symbol transmitted at time pT. Without “loss of”, generally, we suggest setting $D_{-1}=0$ to initiate the integration processing. At the receiver side, the estimation of ${\left\{{\hat{S}}_{p}\right\}}_{p\geq 0}$ is obtained as:(20)$${\hat{S}}_{p}=\left({\hat{D}}_{p}-{\hat{D}}_{p-1}\right)\;\mathrm{mod}\;\;M$$
where the estimation of the DCSS symbols $\left\{{\hat{D}}_{p}\right\}$ are based on the periodogram method presented in Section 2.2. Thus, as expressed in (Equation 14) and (Equation 15), the differential process performed by (Equation 20), limits the impact of $\left(-L+C\right)$ on the symbol estimations.

However, it is necessary to estimate and compensate the fractional CFO ν and the fractional STO λ to prevent performance degradation. Furthermore, to insure high robustness of DCSS to the variation of Doppler shift over time, the latter technique is combined with more precise estimation of the frequencies that maximize the module of the FFTs, as described in the next paragraph.

#### 3.1.2. Additional Processing at the Receiver

In the presence of the time varying Doppler shift, it is judicious to implement more precise estimation of the argmax of each FFT module. To address this, many techniques have been developed in the literature. For instance:Quadratic interpolation;Secant method;Newton’s method;Bisection method.

For more details on the latter methods, the reader can refer to [29]. In this work, we used a low-complexity technique based on the Bisection method. Thus, we consider the interval $\left[a=\omega_{{\hat{D}}_{p}-1},b=\omega_{{\hat{D}}_{p}+1}\right]$, with $\omega_{{\hat{D}}_{p}}=\frac{2\pi ({\hat{D}}_{p}-1)}{M}$ being the pulse that matches the symbol ${\hat{D}}_{p}$, and we maximize the following function:(21)$$R\left(\omega \right)=\left|\sum_{n=0}^{M-1}z\left(n,\;p\right)e^{-j(n-1)\omega }\right|,\;\omega \in \left[a,\;b\right]$$

With z(n, p) defined as in (Equation 16), with the transmitted symbol as ${\hat{D}}_{p}$. We considered a non-coherent demodulation to take advantage of the robustness of DCSS against the phase variation.

We propose to numerically compute an approximation of ${\hat{D}}_{p}$ with an error less than a given maximum permissible error ϕ. A trivial solution is to consider *N* equidistant points $y_{1}=a<y_{2}<\cdots <y_{N}=b$ with $N>\frac{b-a}{\phi }$, to calculate $R\left(y_{1}\right),\;R\left(y_{2}\right),\;\ldots ,\;R\left(y_{N}\right)$ and to find the index of the maximum of this sequence. This method requires the calculation of R(ω) over *N* points. Therefore, it has a complexity in the order of $O\left(M\right)=O\left(\frac{b-a}{\phi }\right)$.

However, taking into account the concave nature of R(ω), the number of points at which the calculation of R(ω) is performed can be significantly reduced by performing a binary search.

The proposed algorithm is as follows:1.Consider a number of points that are of a power of two. More precisely, $p=log_{2}\left(\frac{b-a}{\phi }\right)+1$ and $N=2^{p}$ are taken. The starting analysis interval is $\left[a,\;b\right]=\left[y_{1},\;y_{2^{p}}\right].$2.Estimate R(ω) at the extremities $y_{1}=a$ and $y_{2^{p}}=b$, and also at the two points “in the middle” of the analysis interval, i.e., $y_{2^{p-1}}$ and $y_{2^{p-1}+1}$. If the maximum of R(ω) calculated in these four points is reached for for one of the two extremities of “half” left $\left[y_{1},\;y_{2^{p-1}}\right]$, this interval becomes the new analysis interval, otherwise the new analysis interval will be the “half” right $\left[y_{2^{p-1}+1,\;}y_{2^{p}}\right]$.3.Loop on step 2 by processing the new analysis interval and continue until step 4 criteria is reached.4.After *p* iterations, the extremities of the analysis interval are two points at a distance of $\frac{b-a}{2^{p}}$. The highest value of R(ω) computed from these points is decided to be the sought solution.

The association between the DCSS technique and the latter interpolation method would allow the proposed receiver to have high robustness against the time-varying CFO. To take advantage of this robustness, we propose in the next paragraph an original synchronization algorithm.

### 3.2. Proposed Synchronization Signal Based on the Use of DCSS

The DCSS transmitter is basically similar to the LoRa one, since the same structures of linear chirps are used. However, additional differential processing is implemented before the chirps are generated. This can be easily implemented, which guarantees the cost-effectiveness of our proposed transmitter.

Indeed, in our DCSS transmitter, the preamble and the SFD symbols are no longer needed to estimate *L* and *C*, using the system of two equations as in (Equation 18), since the latter modulation is robust to frequency desynchronization and tolerates some timing misalignment that does not induce important ISI. Therefore, in this work, we propose an original method to estimate time offset regardless of the frequency offset. To this end, we use the $N_{p}$ up-chirps of the preamble for the signal detection and the estimation of the fractional offsets. We also maintain the *sync word* to verify the accuracy of the time synchronization and one down-chirp symbol, as an SFD, to adjust the receiver’s timing alignment. This structure of preamble is very similar to the ones deployed in LoRa PHY layer, except for the number of symbols of the SFD, which is reduced to one symbol.

If we note $x_{pre}\left(t\right)$, the complex envelope of the proposed synchronization signal and s(t), the continuous-time version of (Equation 3) where the transmitted symbols are ${\left\{D_{p}\right\}}_{p\geq 0}$, the signal transmitted by a DCSS node can be written as follows:(22)$$x\left(t\right)=x_{pre}\left(t\right){𝟙}_{\left[0,\;T_{p}\right)}\left(t\right)+s\left(t-T_{p}\right){𝟙}_{\left[T_{p},\;T_{p}+N_{s}\times T\right)}\left(t\right)$$
where $T_{p}$ is the duration of the latter synchronization sequence, which is equal to $(N_{p}+N_{sw}+1)T$. A spectrogram example of the transmitted signal is shown on Figure 2.

### 3.3. Proposed Synchronization Algorithm

To implement our synchronization algorithm, we consider the same model of the received signal as in (Equation 12). To be more general, the global time desynchronization parameter is supposed to be $t_{s}=KT+\Delta \tau$, K∈N. Thus, the continuous version of the received signal is written:  
(23)$$y\left(t\right)=\sqrt{P}x\left(t-t_{s}\right)e^{j(2\pi \left(\Delta f+\frac{c_{d}t}{2}\right)t+\theta_{0})}+w\left(t\right)\;$$

As we already noted, due to the CFO and the DR, the ADC output signal should be sampled with $f_{s}^{\prime }=\alpha f_{s_{min}}$, which will allow a more accurate receiver alignment with the effective start of the payload, and is necessary to correctly capture the power spectral density of the signal to process. Nevertheless, to ensure a low complexity of our proposed receiver and for a fair comparison with LoRa performance, the different processing steps are developed with a sampling rate, such that $T_{s}=\frac{1}{f_{s_{min}}}$. Hence, The received signal, sampled at $T_{s}$, can be written as:(24)$$y\left(n\right)=\sqrt{P}x\left(n-n_{s}\right)e^{j(2\pi \left(\Delta f+\frac{f_{d}\left(nT_{s}\right)}{2}\right)nT_{s}+\theta_{0})}+w\left(n\right)\;$$
where
(25)$$n_{s}=\left\lfloor \frac{t_{s}}{T_{s}}\right\rfloor =KM+\left\lfloor \frac{\Delta \tau }{T_{s}}\right\rfloor$$

DCSS, as well as all the other modulation techniques, do not escape the need for an accurate time synchronization to avoid ISI, which strongly degrades the receiver sensitivity. However, in DCSS as well as in CSS, the presence of a time varying Doppler shift degrades the synchronization and the decoding performance, especially when increasing the symbol time *T*. Therefore, an estimation and compensation of the DR are mandatory in some cases.

Given these properties of DCSS, we propose to perform the synchronization of (Equation 24) by implementing the following six steps, detailed hereafter:1.Preamble detection;2.Coarse time synchronization;3.Doppler rate estimation;4.Fractional CFO estimation;5.Fractional STO estimation;6.DR, fractional CFO, and fractional STO compensations.

#### 3.3.1. Step 1—Preamble Detection

The first step in our synchronization algorithm is the detection of a signal of interest through the search of the known preamble. To this end, the receiver must be in a listening mode, which is done by multiplying each block of *M* samples by the complex conjugate of the reference signal as written in (Equation 13). Then a FFT is calculated on each block of non-overlapping *M* samples, as in (Equation 8). To increase the certainty of the preamble detection, it is advantageous to average the FFT magnitudes of successive blocks before applying the argmax function. Indeed, since in the preamble the symbols are identical, this processing would average out the bin containing the noise, easing finding the correct one. To do this, authors in [14,19] propose designing an IIR filter, such as y[n]=x[n]+αy[n−1] instead of averaging, with α<1 being the portion of the previous block to be remembered. In this work, we chose to average the FFT magnitude over each two consecutive blocks. In addition, as stated in [14], the performances are enhanced if a threshold value, according to the noise level, is set to determine the presence of the signal peaks in the FFTs. Subsequently, a preamble is assumed to be detected when in $\left(N_{p}-1\right)$ blocks of *M* samples, the maximum FFT absolute value is on the same FFT bin. However, due to the fractional STO and CFO, and the presence of a significant DR, the positions of the FFTs argmax would be shifted by several FFT bins from the beginning to the end of the preamble up-chirps. Hence, proper control procedures must be envisaged to take into account all these effects when searching for the preamble. In other words, we do not have to look for $\left(N_{p}-1\right)$ consecutive peaks at the same FFT bin. In the same context, authors in [14,22] propose relaxing this constraint by searching for only $\frac{N_{p}}{2}$ consecutive peaks at the same frequency, with a tolerance up to ±2 FFT bins.

After detecting the presence of a valid preamble, our receiver should identify in which *T*-long sequence the received packet begins. To do this, a sequence of up-chirps was applied to the *T*-long sections where the down-chirp of the SFD is expected. The module of the FFT having the highest maximum indicates the location of the *T*-long section of the SFD. Given the latter position, $\hat{K}$ an estimation of *K* in (Equation 25) can be deduced.

#### 3.3.2. Step 2—Coarse Time Synchronization

Before starting the demodulation process, it is mandatory to be time synchronized at the beginning of the frame to avoid ISI. Nevertheless, thanks to the differential process, the DCSS modulation is more robust than CSS to time synchronization errors. Therefore, a time alignment that ensures a predominant cardinal sine in each FFT is sufficient to achieve accurate decoding performance. Based on this feature, we propose, in this step, to coarsely estimate $n_{s}$, the frame beginning. Indeed, after estimating $\hat{K}$ and considering the distribution of $\frac{\Delta \tau }{T_{s}}$ in the set $\left[-\frac{M}{2},\;\frac{M}{2}\right)$, the signal’s beginning instant $n_{s}$ will be in the range of $⟦a\times M,\;b\times M⟧$ with $a=\hat{K}-\frac{1}{2}$ and $b=\hat{K}+\frac{1}{2}$.

Here, it should be noted that the maximum possible FFT magnitude is obtained in a perfect time alignment. Otherwise, the energy of the main peak will span over several bins and two cardinal sines may appear. Thus, the principle of our coarse time synchronization method is to search for the starting index that maximizes the magnitude of all FFTs in the preamble detected in step 1. Thereby, the function that we propose to maximize can be written as follows:(26)$$H\left(\omega \right)=\sum_{p=\omega }^{\omega +N_{p}+N_{sw}-1}\underset{k}{max}\left(\left|Y[k,\;p]\right|\right),\;\;\omega \in \left[a,\;b\right]$$

To guarantee a symmetric property between H(a) and H(b) (i.e., in a perfect time synchronization, no peaks would appear in the FFT before the preamble and in the one after the *sync word*), the down-chirp of the SFD is inserted before the beginning of the payload. Moreover, a silence period can be considered instead of the SFD. However, the use of latter SFD is mandatory since it is also used in the estimation of *K* as presented in the previous paragraph. Indeed, this SFD can be seen as a guard interval since up- and down-chirps are orthogonal. Finally, ${\hat{n}}_{s}$ the estimate of $n_{s}$, is obtained by searching the index that maximizes the function H(ω) as explained in the pseudo-code of Algorithm 1.
**Algorithm 1:** Proposed estimation of $n_{s}$.
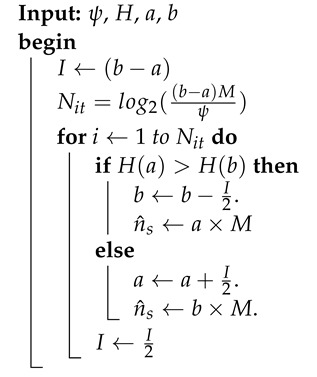


To reduce the computational complexity of this step, we suggest implementing the maximum research by dichotomy. Furthermore, to estimate the start index of the received frame, we have to set the maximum permissible error of the Algorithm 1 to ψ=1 sample, which gives a number of iterations $N_{it}=log_{2}\left(M\right)=SF$.

If we note r(n), the coarse time synchronized signal, we have:(27)$$r\left(n\right)=y\left(n+{\hat{n}}_{s}\right),\;\;n\in I_{1}$$
with $I_{1}=\left\{0,\;\ldots ,\;\left(N_{p}+N_{sw}+1+N_{s}-1\right)M-1\right\}$.

#### 3.3.3. Step 3—Doppler Rate Estimation

After performing the time synchronization, the receiver has to estimate the DR to remove the impact of the time varying frequency shift $f_{d}\left(t\right)$ on symbol estimation. To this end, we propose an algorithm based on the estimation of the peak position in each *T*-long sequence of the preamble up-chirps. The peak frequency values are processed in order to find the linear regression, which represents the frequency slope due to the DR. The proposed algorithm, using the same principle as in [22], is summarized by the following three points:(i)Estimate the argmax of the FFT module in each symbol interval of the preamble. If we note $i_{p}$, the argmax of the FFT module in the $p^{th}$
*T*-long sequence, we have:
(28)$${\hat{i}}_{p}=\underset{k\in ⟦0,\;M-1⟧}{argmax}\left(|R\left[k,\;p\right]|\right)$$
with $R\left[k,\;p\right]=\frac{1}{\sqrt{M}}\sum_{n=0}^{M-1}\left(r\left(n,\;p\right)x_{ref}^{*}\left(n\right)\right)e^{-j2\pi \frac{nk}{M}}$ and r(n, p)=r(n+(p−1)M), $\forall n\;\in \;⟦0,\;M-1⟧$. It should be noted here that the interpolation method, as presented in Section 3.1.2, is used to increase the accuracy of the estimate ${\hat{i}}_{p}$, while in [22], a classical argmax function, is performed.(ii)The FFT argmax is used in pairs to compute different DR estimates noted ${\hat{c}}_{d}^{p,l}$. These estimations are obtained using the couple $\left\{{\hat{i}}_{p},\;{\hat{i}}_{p+l}\right\}$, with $p\;\in \;\{1,\;N_{p}-1\}$ and $l\;\in \;\{p+1,\;N_{p}\}$. Thus, by considering (Equation 17), we have:
(29)$${\hat{c}}_{d}^{p,\;l}=\frac{2}{T^{2}}\frac{{\hat{i}}_{p+l}-{\hat{i}}_{p}}{l}$$(iii)An estimation of the DR is obtained by averaging ${\hat{c}}_{d}^{p,\;l}$, as follows:
(30)$${\hat{c}}_{d}=\frac{1}{N_{p}-1}\sum_{p=1}^{N_{p}-1}\frac{1}{N_{p}-p}\sum_{l=p+1}^{N_{p}}{\hat{c}}_{d}^{p,l}$$

We note that the estimation of the DR is done at the sampling rate $T_{s}=\frac{1}{f_{min}}$, while in [22], the sampling rate is equal to $\frac{1}{2f_{min}}$. Furthermore, this estimation is needed only if the frequency separation between two adjacent bins, $\Delta_{b}=\frac{1}{T}=\frac{B}{M}$ is greater than a specific value. In this case, the compensation of the DR is mandatory before starting the estimation of the fractional offsets. In the payload, the compensation of the DR is done after the compensation of the fractional STO and perform the downsampling at the frequency rate $f_{min}$. In the next section, the robustness of our proposed receiver is tested with different separation $\Delta_{b}$, i.e., different values of *B* and SF.

#### 3.3.4. Step 4—Fractional CFO Estimation

The compensation of the fractional CFO ν is mandatory to avoid off-by-one demodulation errors and the degradation of the SNR after the dechirping process. One method to estimate ν was described in [27] using a variant of the well-known Schmidl–Cox estimator [30]. This estimator averages the phase differences between samples with the same index from two consecutive chirps carrying the same symbol. Given that the transmitted signal starts with $N_{p}$ unmodulated up-chirps of the preamble, an estimate $\hat{\nu }$ of ν is obtained, after compensating the DR, as follows:(31)$$\hat{\nu }=\frac{1}{N_{p}-1}\sum_{p=1}^{N_{p}-1}\frac{1}{2\pi }\arg\left(\sum_{n=0}^{M-1}r_{c_{d}}\left(n,\;p\right)r_{c_{d}}^{*}\left(n,\;p+1\right)\right)$$
where $r_{c_{d}}\left(n,\;p\right)\;=\;r\left(n\;+\;\left(p-1\right)M\right)e^{-j\pi {\hat{c}}_{d}n^{2}T_{s}^{2}}$, $\forall n\;\in \;⟦0,\;M-1⟧$. Here, we note that the compensation of the time varying Doppler is mandatory to prevent the changing of the fractional CFO from one symbol to another.

#### 3.3.5. Step 5—Fractional STO Estimation

To achieve an accurate receiver alignment, it is necessary to compensate the fractional STO λ. Indeed, once the fractional CFO and the DR are estimated, the latter are compensated in the unmodulated up-chirps of the preamble to allow the receiver to perform the estimation of λ. To this end, we propose to compute the following FFT, denoted $R_{c_{d},\;\nu }\left[k,\;p\right]$, in the $p^{th}$
*T*-long sequence of the preamble after the compensation of $\hat{\nu }$ and ${\hat{c}}_{d}$:(32)$$R_{c_{d},\nu }\left[k,\;p\right]=\frac{1}{\sqrt{M}}\sum_{n=0}^{M-1}\left(r_{c_{d}}\left(n,\;p\right)x_{ref}^{*}\left(n\right)e^{-j2\pi \frac{n\hat{\nu }}{M}}\right)e^{-j2\pi \frac{nk}{M}}$$

We notice in (Equation 32) that residual errors in estimating the values of ν and $c_{d}$ would impact the estimation process of λ.

To compute the fractional STO in the latter *T*-long sequence, an interpolation method can be used to find a finest estimation of the frequency that maximizes $|R_{c_{d},\nu }\left[k,\;p\right]|$. We note ${\hat{\lambda }}_{p}$ this estimate. Once again, and to be compliant with the low power constraints of LPWAN, we propose a low complexity method to accurately find ${\hat{\lambda }}_{p}$. This method is based on the maximization of concave functions by dichotomy. It should be noted that LPWANs attempt to offload complexity from the end devices, as terrestrial gateways, do not have power consumption constraints. However, when implementing onboard decoding on the satellite, the complexity of the algorithms must be taken into account.

To obtain a precise estimation of λ, an averaging over the preamble up-chirps is performed as follows:(33)$$\hat{\lambda }=\frac{1}{N_{p}}\sum_{p=1}^{N_{p}}{\hat{\lambda }}_{p}$$

#### 3.3.6. Step 6—DR, Fractional CFO, and Fractional STO Compensation

The DR is compensated in the preamble to allow accurate estimation of ν and λ. In the payload, the receiver has to compensate fractional STO, perform the down-sampling at $f_{s_{min}}$, and then compensate the DR and ν.

To this end, based on the estimate $\hat{\lambda }$, the timing alignment is done in the decimation chain of the receiver’s digital front-end. Before the samples are produced at the minimum sampling frequency $f_{s_{min}}$, an oversampling $f_{s}^{\prime }=\alpha \times f_{s_{min}}$ is considered. Indeed, the compensation of the fractional STO $\left\lfloor \alpha \times \hat{\lambda }\right\rceil$ can be easily done by shifting the decimation operator’s input by a corresponding number of undecimated samples. After that, the payload is sub-sampled at the frequency rate $f_{s_{min}}$. Thus, the symbols can be easily estimated as depicted in (Equation 10), where the FFTs are computed as in (Equation 32) after the compensation of $\hat{\nu }$ and the DR.

Finally, before starting the payload decoding, the receiver has to verify the accuracy of our synchronization algorithm by finding the two special modulated symbols of the *sync word*, as depicted in Figure 3, which summarizes the architecture of our proposed receiver.

In the next section, we provide simulation results to demonstrate the efficiency of our DCSS receiver.

## 4. Results and Discussion

In this section, we evaluate the performance of our proposed receiver in performing accurate synchronization and decoding of the received DCSS signals. To this end, we perform Monte Carlo-based simulations, with a significant number of repetitions per SNR, using synthesized LoRa-like signals. The simulation results we present are obtained from a DCSS signal simulator that we developed in MATLAB. Thus, we simulated the interleaving and de-interleaving blocks, but also the channel coding/decoding parts, recalled hereafter. Our first test consisted of evaluating the robustness of DCSS technology against the CFO, STO, and the DR. Based on the latter robustness, especially against the significant Doppler variation, we then show the ability of this waveform associated with our original synchronization algorithm to demodulate LEO satellite signals.

In the following, we consider in all the simulations:A number of preamble up-chirps $N_{p}=8$;A reference bandwidth $B_{ref}=125$ kHz, which is the most commonly used bandwidth in LoRa-based networks;An oversampling factor α=8, which gives a typical sampling frequency in LoRa receivers $f_{s}^{\prime }=1$ MHz, when the bandwidth of the signal is equal to $B_{ref}$;Δτ (respectively, Δf) uniformly distributed in $\left[-\frac{T}{2},\;\frac{T}{2}\right]$ (respectively, $\left[-\Delta f_{max},\;\Delta f_{max}\right]$).

### 4.1. Channel Coding and Interleaving in LoRa

In order to increase the robustness of the LoRa modulation against interfering bursts and off-by-one demodulation errors, bits were encoded before the chirp generation. The encoding stages are as follows:

#### 4.1.1. Interleaving

Interleaving is a process that scrambles data bits throughout the packet. It is often combined with forward error correction (FEC) to make the data more resilient to bursts of interference [31]. According to the patent [13], a diagonal interleaver is implemented in LoRa chips.

#### 4.1.2. Forward Error Correction

FEC is used for controlling errors in data transmission over unreliable or noisy communication channels. In LoRa, Hamming FEC is used with a variable codeword size ranging from 5 to 8 bits [13]. Furthermore, the data size per codeword is set to 4 bits, which allow defining the coding rate as $\frac{4}{4+CR}$, where CR∈{1,…4} is the code rate or is also the number of redundancy bits.

### 4.2. DCSS Performance Evaluation and Comparison with CSS

It should be noted that this comparison between DCSS and CSS is done without implementing the aforementioned channel coding. The FEC and the interleaving will be deployed only when presenting the performance of our receiver.

Given the principle of DCSS as described in Section 3.1, one can remark that this modulation naturally introduces a degradation of performance compared to CSS. Indeed, two consecutive DCSS symbols must be properly detected for the original symbol carrying the information to be accurately retrieved (see (Equation 20)). However, Figure 4 proves that this impairment remains low. For example, if a bit error rate (BER) equals $10^{-4}$, the loss is only of 0.2 dB for all of the SFs.

The second evaluation test of DCSS is to assess its robustness to a linear time varying Doppler shift. To this end, we represent in Figure 5, the packet error rate (PER) of the latter modulation as function of the DR in a perfect synchronization case (i.e., Δτ, Δf=0) and without considering the noise. To note that the DR is not compensated in these simulations. It can be seen that the decrease of the maximum permissible error of the dichotomy method ϕ leads to an enhancement of the robustness of DCSS against the DR. However, this increase of precision is at the expense of the computational complexity. Indeed, for $\phi =\frac{1}{M}$, the DR limit that can be naturally supported by DCSS is $c_{d}^{th}=385$ Hz/s. For $\phi <\frac{1}{M}$, the robustness of DCSS is clearly degraded and for $\phi >\frac{1}{M}$, the gain is not important. Based on this simulation, we can confirm that $\phi =\frac{1}{M}$ is our optimal choice when using the bandwidth $B_{ref}$. In general, for any bandwidth configuration, the precision parameter ϕ should be equal to $\frac{\Delta_{b}}{B_{ref}}$, to have the same performance as in Figure 5.

To compare the robustness of DCSS and CSS to Doppler time variation, we represent, in Table 1, the DR limit that can be supported by each waveform, with different SFs at the bandwidth $B=B_{ref}$ and $\phi =\frac{1}{M}$. To ensure the fairness between the two physical layers, the DR is not compensated in both cases and simulations are done in a perfect scenario (i.e., perfect synchronization without the AWGN channel).

It can be seen in the above table that DCSS associated with the interpolation technique, as described in Section 3.1, is much more robust to DR than CSS. This result is explained by the capacity of the differential process combined with the interpolation to retrieve the effective symbols in the presence of fractional offsets when estimating the frequencies of the main peaks in the FFTs. In LoRa, the received signal can be accurately decoded only if the variation, cased by the DR, between each consecutive symbols is lower than the half of the frequency separation between each two adjacent bins, which is not the case in the DCSS technique.

We note that the latter results are not dependent on the SF only, but also on the bandwidth *B*. In other terms, the frequency separation $\Delta_{b}$ is the parameter that defines the robustness of DCSS technology to DR. For instance, the robustness to DR of SF=12 with $B=B_{ref}$ is the same as SF=7 with $B=\frac{B_{ref}}{2^{5}}$.

Finally, we represent in Figure 6 the BER evolution of DCSS and CSS for SF=12, $B=B_{ref}$ and a payload size $N_{pay}=51$ bytes (According to LoRaWAN protocol, the maximum payload size for the slowest data rates, SF∈{10, 11, 12} on 125 kHz is 51 bytes), for different DR values, in an AWGN channel model. To ensure the fairness between the latter waveforms, the DR is not compensated for both cases and we assume that Δf and Δtau are equal to zero.

The results confirm the fact that DCSS is much more robust to DR than CSS. For instance, it can be easily seen that, for CSS signals, the PER is greater than 0.5 for all SNR values, if the DR is equal or greater than 12 Hz/s. Whereas, for DCSS signals, a PER equal to $10^{-3}$ (respectively, $10^{-2}$) is achieved at SNR =−18 dB for DR =200 Hz/s (respectively, DR =240 Hz/s).

After evaluating the performance of DCSS, we dedicate the next paragraph to assessing the robustness of our proposed receiver when communication with LEO satellites is considered.

### 4.3. Evaluating the Proposed Receiver with LEO Satellite Communication

In this section, we use data provided by Eutelsat https://www.eutelsat.com/en/satellites/leo-fleet.html, accessed on 2 June 2021, a company specializing in the deployment of LEO satellites.

In order to evaluate the performance of our receiver, we first present, in Figure 7, the variation of the DR and Doppler shift over time obtained from an Eutelsat nano-satellite with a typical altitude of 550 km and given the carrier frequency of the LoRa signals 868 MHz.

It can be seen that, in the worst case, the DR can reach 280 Hz/s. We note here that, along the packet duration, the DR can be modeled as a linear shift variable in time. Whereas the Doppler shift can achieve 19 kHz. This significant Doppler shift, related to the satellite motion, combined with local oscillator instability, leads to huge CFO values. Therefore, our proposal, which allows decoding LoRa-like signals whatever the frequency offset, would be a very promising solution for ultra narrow band (UNB) communications with LEO satellite using chirped signals.

As depicted in Table 2, our receiver has to deal with a significant CFO value (i.e., $\Delta f_{max}>\frac{B_{ref}}{4}$) and the fastest Doppler variation in LEO satellite communication.

We tested our algorithm with the (almost) worst case scenario, in terms of the value of DR in LEO communications with altitude ranging between 300 and 700 km, such as the majority of CubeSats [32]. Moreover, the values of DRs and Doppler shifts depend on the carrier frequency, and since there were not a lot of works that implemented LEO satellites communications using the the 868 MHz carrier frequency, we did not find sufficient results in this context in the literature.

#### 4.3.1. Synchronization Algorithm Numerical Results

In the following simulations, we consider the worst case scenario of LEO satellite communications, as proposed in Table 2.

Figure 8 shows the estimation error of the fractional CFO $\epsilon_{\nu }=\left|\nu -\hat{\nu }\right|$ for all possible SFs. It can be seen that the estimation of ν is more precise for the lower SF (i.e., SF∈{7,8,9}) since a DR of 280 Hz/s does not affect the synchronization and the decoding performance of the latter SFs. In addition, the highest SFs have the lowest bin separations, which make them more sensitive to the fractional offsets and the DR.

In Figure 9, we represent the start of frame error $\epsilon_{n_{s}}=\left|\frac{n_{s}+\lambda -\left({\hat{n}}_{s}+\hat{\lambda }\right)}{M}\right|$ as a function of the SNR for each SF. We notice that our time synchronization algorithm has good precision since, for instance, $\epsilon_{n_{s}}=0.017$ (respectively, $\epsilon_{n_{s}}=0.021$) at the SNR sensitivity threshold (the SNR associated to a BER of $10^{-5}$ in LoRa communication) [15] of SF=7 (respectively, SF=12). In [21], we showed that DCSS can maintain good decoding performance for timing errors $\epsilon_{n_{s}}$ of 0.25. Hence, given the time synchronization accuracy of our algorithm and the robustness of DCSS, we expect to have good decoding performance of our receiver.

#### 4.3.2. Decoding Performance of the Proposed Receiver

After evaluating the performance of our receiver to estimate the parameters needed to perform an accurate synchronization, we present in Figure 10, a comparison of the decoding performance of our receiver and a classic LoRa one as function of the static CFO (i.e., without the DR) with a CR=1. The FEC with CR=1 only allows detecting the presence of errors and does not correct them. We used a CR>1 in the upcoming simulations in order to reduce the impact of off-by-one demodulation errors caused by the residual fractional offsets, especially the DR.

In this figure, we represent the PER with a number of transmitted packets equal to $10^{4}$ and SNR equal to −5 dB. This simulation confirms the constraint of a maximum allowed CFO of $\frac{B}{4}$ for LoRa receivers, which is not the case when adopting our approach.

In Figure 6, we compare the robustness of DCSS and CSS waveforms against the DR under the AWGN channel. In Figure 11, we propose the same comparison test between our DCSS receiver and CSS one, as described in [27], in the presence of the STO and CFO, but without compensating the DR. In this simulation, we used the configuration (SF=9, $B=\frac{B_{ref}}{2^{3}}$), which has the same robustness to the DR as (SF=12, $B=B_{ref}$). We notice in this figure that the CSS receiver is very sensitive to the Doppler variation since, for a DR = 10 Hz/s, an almost constant PER of 0.5 is obtained for SNRs greater than −13 dB. On the other hand, the DCSS receiver maintains an acceptable decoding performance for DR values lower than 70 Hz/s. For instance, for a PER = $10^{-3}$, our receiver has a loss of SNR of only 2.5 dB with a DR = 70 Hz/s compared to the perfect synchronization case. These results are consistent with those in Figure 6. However, we notice that the robustness of both receivers to DR in the presence of the CFO and the STO are lower than the perfect synchronization case as presented in Figure 6, since an uncompensated DR would affect the estimation of the latter desynchronization parameters.

Figure 12 states the results of PER of our proposed receiver as a function of the SNR for all SFs. We consider the worst case of DR and the maximum of payload size with each SF as defined by the LoRaWAN standard in [33]. Thanks to the accuracy of our synchronization algorithm and the robustness of the DCSS technique to CFO and some STO values, we notice in Figure 12 that the decoding performance of our receiver is slightly degraded compared to the perfect synchronization case. It can be seen that PER is slightly increased for the lowest bin separation $\Delta_{b}$ (i.e., slowest data rates, SF∈{10,11,12}). This result is explained by a higher sensitivity of the latter SFs to the time-varying Doppler shift and the fractional CFO. We also notice that the performance degradation, compared to the perfect synchronization case of the configuration SF=12 and $B=B_{ref}$, is almost the same than SF=7 and $B=\frac{B_{ref}}{2^{5}}$, since they have the same bin separations. It should also be noted that these two configurations have the same link budgets.

Finally, in Figure 13 we present the SNR evolution of an uplink line of sight communication between the Eutelsat satellite and a terminal in its field of view (FoV) as a function of time. We also show the elevation angle from the terminal to the satellite during the visibility window [34]. Let $d_{u}\left(t\right)$ be the distance between the satellite and the user device, which depends on the elevation angle of the satellite during its window of visibility. The SNR acquisitions, as shown in the latter figure and (Equation 34), are obtained by considering an omnidirectional transmitter antenna having a gain $G_{Tx}=0$ dBi, a directional receiver antenna with a gain $G_{Rx}=8$ dBi, and a polarization mismatch $L_{P}=-3$ dB.
(34)$$\begin{array}{cc} \mathrm{SNR}{|}_{\mathrm{dB}} & =P_{Tx}{|}_{\mathrm{dBm}}+G_{Tx}{|}_{\mathrm{dBi}}+L{|}_{\mathrm{dB}}+L_{P}{|}_{\mathrm{dB}}\\ & +G_{Rx}{|}_{\mathrm{dBi}}-10\;\log_{10}\left(k_{B}T_{N}B\right)-30 \end{array}$$
where $k_{B}=1.38\times 10^{-23}$ J/K is the Boltzmann constant, $T_{N}=298,\;15$ K is the equivalent noise temperature, $L{|}_{\mathrm{dB}}=20\log_{10}\left(\frac{c}{4\pi f_{c}d_{u}\left(t\right)}\right)$ is the line of sight path loss, and *c* is the speed of light.

By referring to the curves of the PER as functions of the SNR in Figure 12, we propose defining the SNR sensitivity threshold ${\mathrm{SNR}}_{th}$ as the minimum SNR that guarantees a PER lower than $10^{-2}$. Using the latter SNR threshold value, we can easily compute the sensitivity of our receiver as follows:(35)$$S{|}_{\mathrm{dBm}}={\mathrm{SNR}}_{th}{|}_{\mathrm{dB}}-174+10\log_{10}\left(B\right)+NF{|}_{\mathrm{dB}}$$
with NF being the noise figure of the receiver and it is equal to 6 dB.

At the farthest distance $d_{max}$ between the satellite and the terminal device (i.e., $d_{u}\left(t\right)=d_{max}$ at the elevation angle of 20${}^{\circ }$), the measured SNR is equal to −19 dB. The latter measurement gives a received power of −136 dBm. In Table 3, based on Figure 12 and (Equation 35), we present the sensitivity of our proposed receiver for each SF and *B* configuration. The results of the latter table prove that only SF=12 with $B=B_{ref}$ and SF=7 with $B=\frac{B_{ref}}{2^{5}}$ can fulfill the sensitivity requirements for all the SNR measurements by the Eutelsat satellite. Hence, any transmitted signal has the same bin separation as $\Delta_{b}=\frac{B_{ref}}{2^{12}}$ and would be suited for this communication. It should be noted that an adaptive data rate communication, according to the position of the satellite, could be considered.

## 5. Conclusions

CSS signals are extremely sensitive to time and frequency offsets, especially the Doppler shift variable in time. In this paper, we proposed a new LoRa-like receiver, to improve the robustness of symbol decoding to synchronization errors. This robustness was obtained by implementing differential symbol coding that modulates the transmitted chirps associated with an original synchronization algorithm. The latter differential processing can be easily implemented, which guarantees the cost-effectiveness of the transmitter. Subsequently, this novel approach allows synchronization of LoRa-like signals by decoupling the estimation of the CFO and the STO, which releases the constraint of the maximum allowed CFO of $\frac{B}{4}$. In addition, our proposed technique is more robust to the time varying Doppler shift than the existing LoRa-based receivers. Simulation results show the efficiency of our receiver in dealing with time and frequency offsets, especially the time variant frequency shift caused by the Doppler effect. Finally, the capacity of our receiver in processing CFOs greater than $\frac{B}{4}$, and its robustness to the DR, make it possible to consider, if the communication rate allows it, UNB communications with LEO satellites using LoRa-like signals. In future work, we plan to evaluate the performance of our proposed receiver in real-time communications using software defined radios. We intend to combine our algorithm that deals with the destructive collision in LoRa, as presented in [8], with the one proposed in this paper.

## Figures and Tables

**Figure 1 sensors-22-01830-f001:**
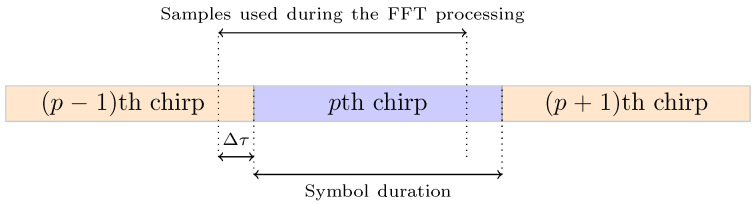
Time desynchronization illustration.

**Figure 2 sensors-22-01830-f002:**
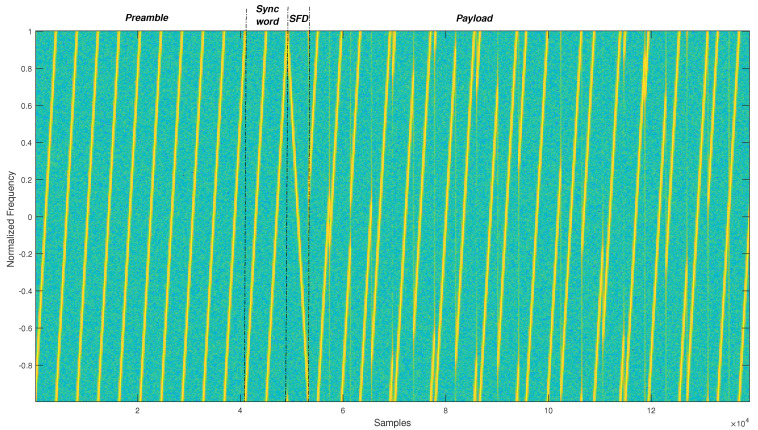
Spectrogram of the transmitted signal.

**Figure 3 sensors-22-01830-f003:**
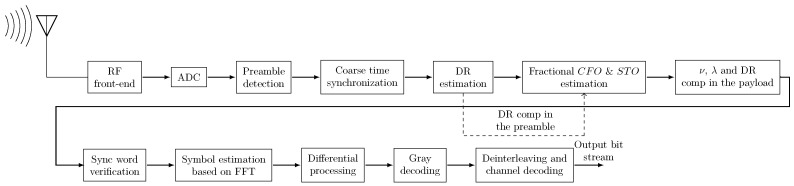
Proposed receiver architecture.

**Figure 4 sensors-22-01830-f004:**
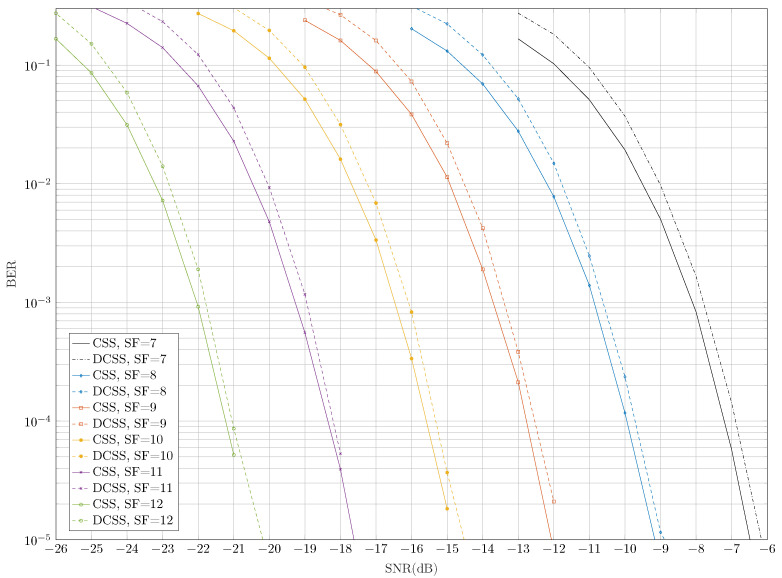
Comparison of bit error probabilities of LoRa and DCSS technologies before channel decoding.

**Figure 5 sensors-22-01830-f005:**
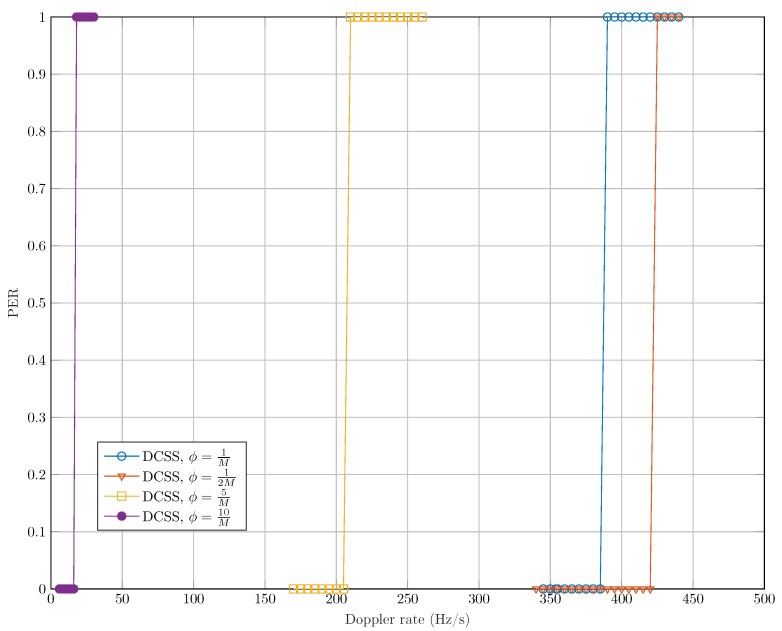
Robustness of DCSS against a Doppler rate with a different complexity order (SF=12, $B=B_{ref}$ and $N_{pay}=51$ bytes).

**Figure 6 sensors-22-01830-f006:**
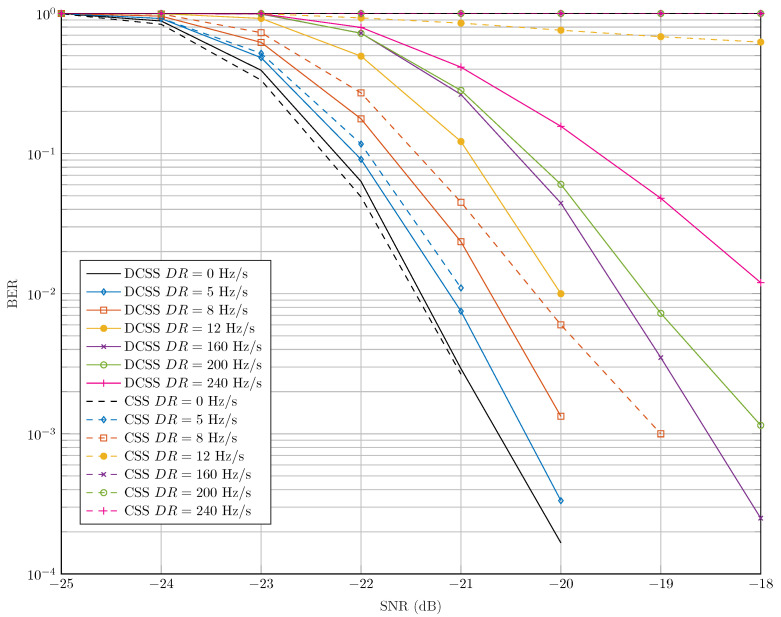
Comparison of robustness of CSS and DCSS technologies to DR over AWGN channel with SF=12, $B=B_{ref}$ and $N_{pay}=51$ bytes.

**Figure 7 sensors-22-01830-f007:**
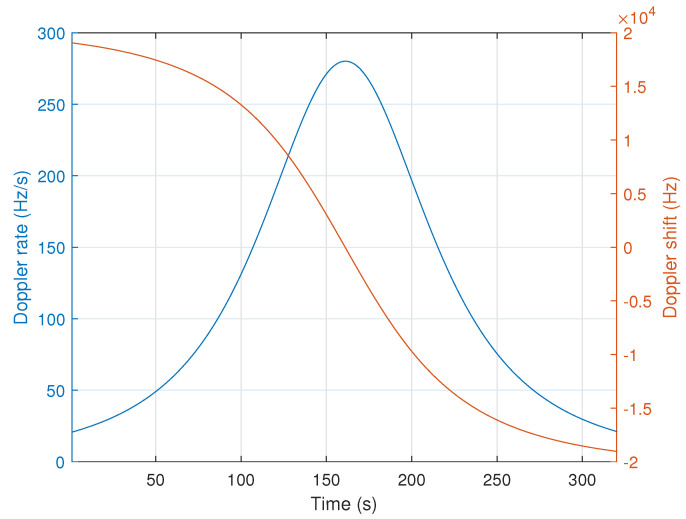
Doppler shift and DR evolution from an Eutelsat nano-satellite with a typical altitude of 550 km and given the carrier frequency of 868 MHz.

**Figure 8 sensors-22-01830-f008:**
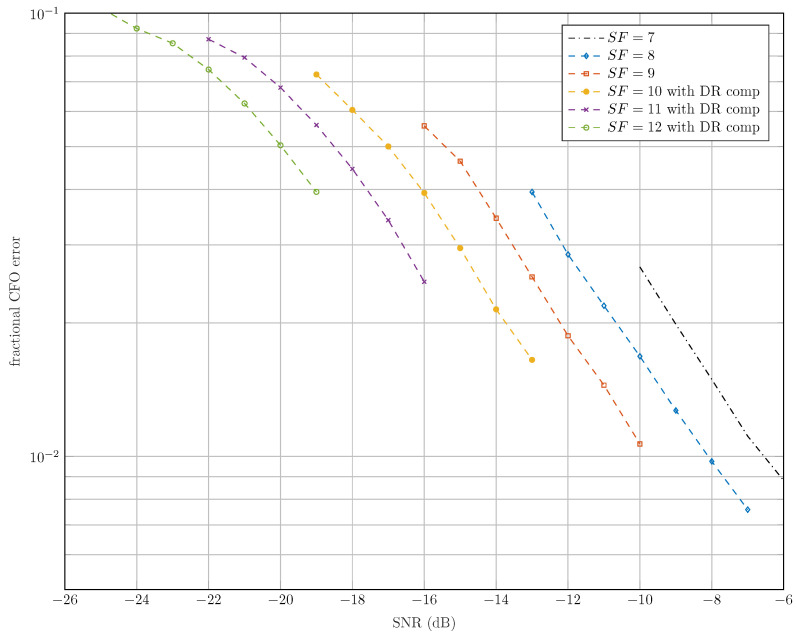
Fractional CFO estimation error $\epsilon_{\nu }$.

**Figure 9 sensors-22-01830-f009:**
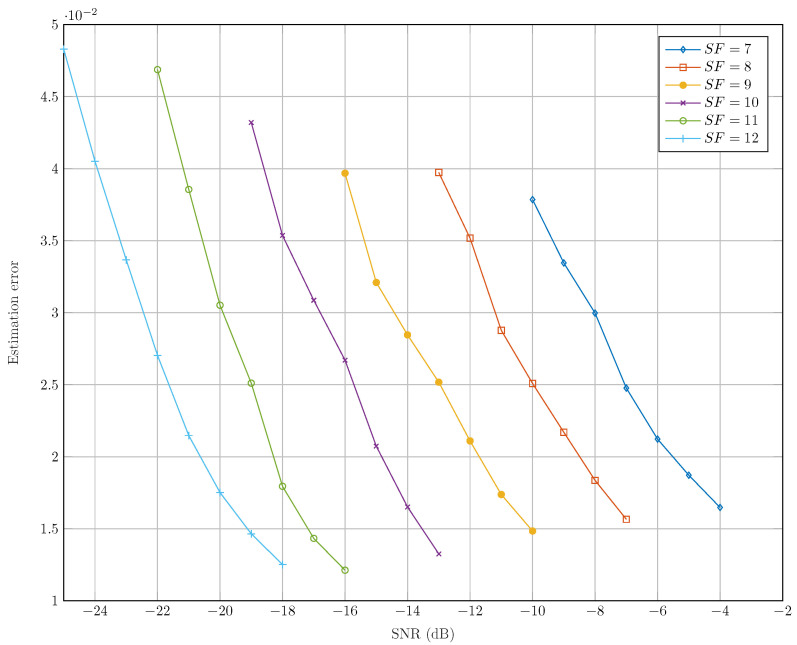
Beginning time estimation $\epsilon_{n_{s}}$.

**Figure 10 sensors-22-01830-f010:**
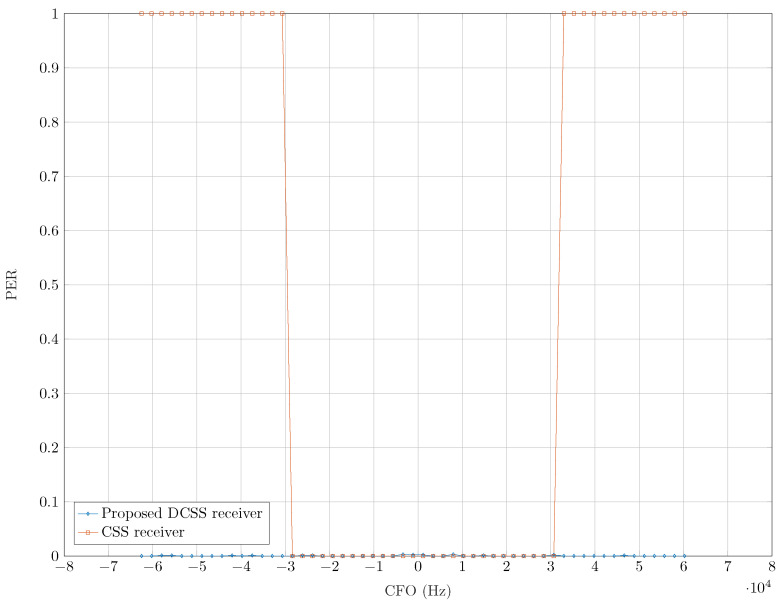
PER of CSS and DCSS receivers as functions of the CFO, with SF=7, SNR =−5 dB, $B=B_{ref}$ and CR=1.

**Figure 11 sensors-22-01830-f011:**
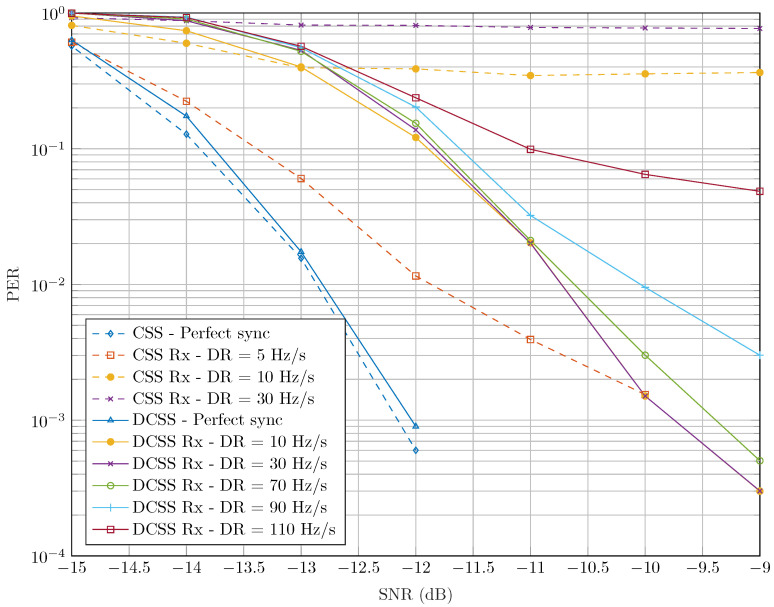
PER of CSS and DCSS receivers for different DR values, which were not compensated, with SF=9, $B=\frac{B_{ref}}{2^{3}}$, CR=1 and $N_{pay}=51$ bytes.

**Figure 12 sensors-22-01830-f012:**
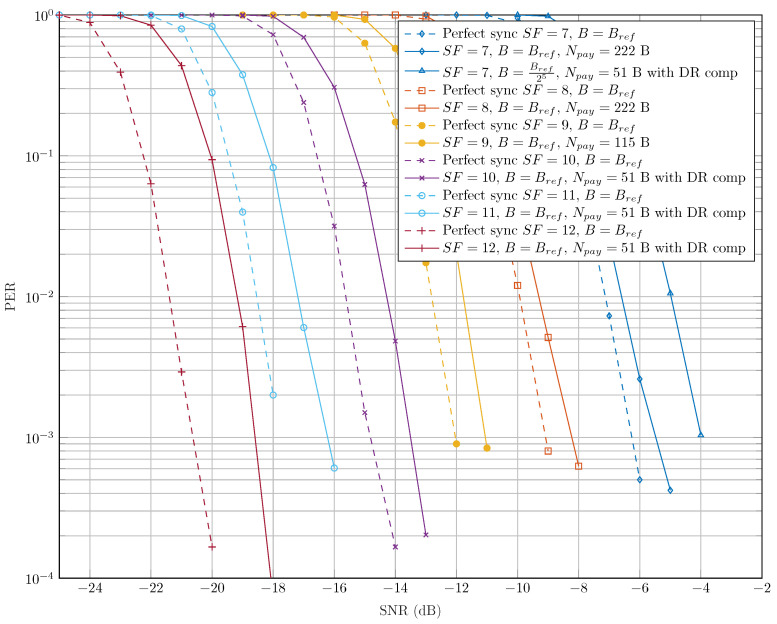
PER evolution of the proposed DCSS receiver with CR=3.

**Figure 13 sensors-22-01830-f013:**
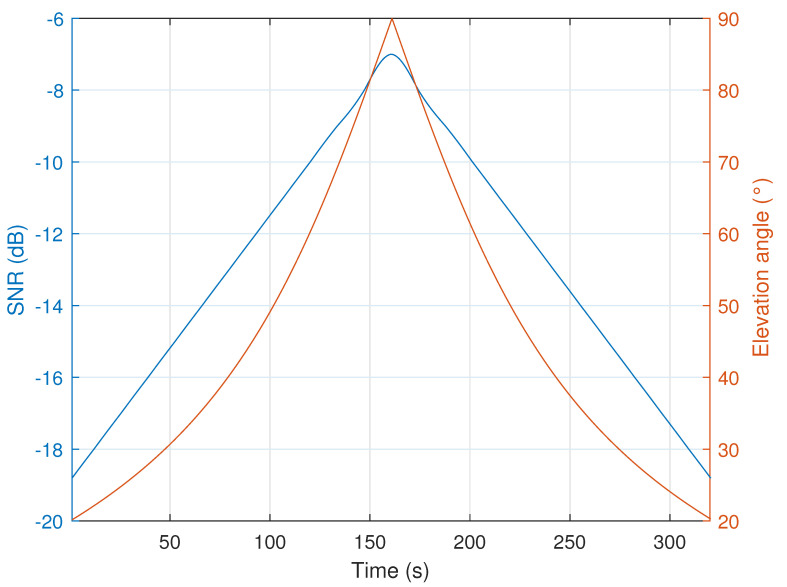
Evolution of the SNR (dB) and elevation angle as function of time for Eutelsat satellite.

**Table 1 sensors-22-01830-t001:** DR thresholds ($c_{d}^{th}$ Hz/s) of DCSS and CSS, B=125 kHz.

SF	7	8	9	10	11	12
Bin separation $\Delta_{b}=\frac{B}{2^{SF}}$ (Hz)	976.56	488.28	244.14	122.07	61.03	30.51
DCSS	394,235	100,605	25,150	6260	1600	385
CSS	9585	2664	713	192	50	13

**Table 2 sensors-22-01830-t002:** Simulation parameters.

Carrier frequency $f_{c}$ (MHz)	868
Maximum CFO $\Delta f_{max}$ (kHz)	50
DR (Hz/s)	280
Transmitted power $P_{Tx}$ (dBm)	14

**Table 3 sensors-22-01830-t003:** Receiver sensitivity *S* dBm for different SF and *B* values.

(SF, B)	($12,\;B_{\mathrm{ref}}$)	($11,\;B_{\mathrm{ref}}$)	($10,\;B_{\mathrm{ref}}$)	($9,\;B_{\mathrm{ref}}$)	($8,\;B_{\mathrm{ref}}$)	($7,\;B_{\mathrm{ref}}$)	($7,\;\frac{B_{\mathrm{ref}}}{2^{5}}$)
$SNR_{th}$ (dB)	−19.5	−17.5	−14.5	−11.8	−9.2	−6.5	−5
*S* (dBm)	−136.53	−134.53	−131.53	−128.83	−126.23	−123.53	−137.08

## Data Availability

Not applicable.

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
