# Peer review of "A New LoRa-like Transceiver Suited for LEO Satellite Communicationsâ€"

_sensors, 2022, doi:10.3390/s22051830_

Round 1
Reviewer 1 Report
- There is a lot of literature on the LoRA modulation and coding and the authors do not refer to any literature when explaining LoRa transmissions. Some of the very critical citaitions that need to be cited:
- https://dl.acm.org/doi/10.1145/3293534
- IoT Network Design using Open-Source LoRa Coverage Emulator
- https://ieeexplore.ieee.org/document/7797659
- One of the important limitations of LoRa is interference, therefore it is essentialt that the authors address this issue which is ten times worse in satellite systems.
- Can the authors comment on how their method would work with coherent demodulation?
- Doppler shift corresponds to a certain satellite height, perhaps the authors can detail what the bounds of the constellation altitudes might be by using their new modulation technique.
Author Response
Reviewer 1 R: There is a lot of literature on the LoRA modulation and coding and the authors do not refer to any literature when explaining LoRa transmissions.Some of the very critical citations that need to be cited:https://dl.acm.org/doi/10.1145/3293534IoT Network Design using Open-Source LoRa Coverage Emulatorhttps://ieeexplore.ieee.org/document/7797659 A: We thank the reviewer for his comment and suggestions. The aforementioned citations are very relevant, we cited them in our manuscript. Thus, the following phrase was added in the introduction: “For instance, several works [1–3] provided detailed studies on the capacity of the latter technology to cope with the requirements of LPWAN ground-based communications, such as the long-range, low energy consumption, and interference resilience.” R: One of the important limitations of LoRa is interference, therefore it is essential that the authors address this issue which is ten times worse in satellite systems. A: We thank the reviewer for his suggestions. He has the right to point out the importance of the interference issue when considering LoRa communication with a LEO satellite. Indeed, in such communications, the interference is a severe problem, due to the field of view of the latter satellite, which allows connecting a huge number of end-nodes. Also, the uncoordinated access to the radio channel as deployed by such LPWAN technology makes the occurrence of this interference very likely. In fact, we dealt with this issue in several of our published papers, such as “An enhanced receiver to decode superposed LoRa-like signals ” and “An enhanced LoRa-like receiver for the simultaneous reception of two interfering signals”. However, we will mention in our manuscript that as future works we plan to combine our algorithm to deal with destructive collisions in LoRa with the one presented in this work. Subsequently, we added the following paragraph in the introduction: “Satellite IoT entails also a higher interference level compared to ground-based IoT, due to the field of views of the latter satellites, which allow connecting a huge number of end-nodes. This issue is aggravated by the uncoordinated access to the radio channel of LoRa-based networks. We dealt with this problem in several of our previous works. For instance, we provided in [4] an approach to deal with destructive collisions in LoRa. In this paper, we will focus only on the synchronization issues when considering LEO satellite communications.” We mentioned also in the conclusion the following statement: “We intend as well to combine our algorithm of dealing with a destructive collision in LoRa as presented in [4] with the one proposed in this paper.” R: Can the authors comment on how their method would work with coherent demodulation? A: We thank the reviewer for his relevant comment. To implement the coherent demodulation, it is mandatory to estimate the initial phase. However, our proposition is based on the differential process of DCSS, which allows being insensitive to the phase variation. Hence, it is relevant to deploy non-coherent demodulation to take advantage of the robustness of DCSS against the phase variation. Hence, we added the following phrase in the DCSS section: “We notice here that we considered a non-coherent demodulation to take advantage of the robustness of DCSSagainst the phase variation.” R: Doppler shift corresponds to a certain satellite height, perhaps the authors can detail what the bounds of the constellation altitudes might be by using their new modulation technique. A: We thank the reviewer for his comment, which is relevant since the Doppler effects depend to the altitude of the LEO satellite. In fact, the speed of a LEO satellite is inversely proportional to its altitude, which means that the Doppler shift decreases if we increase the latter altitude. In our case, we considered a satellite having an altitude of 550 km, which is relatively low. Hence, the tested values of Doppler shifts are very important. Also, it is important to mention that our approach has good robustness against the carrier frequency offset (CFO) and allows us to bypass the constraint of maximum CFO estimable of the quarter of the bandwidth. Theoretically, our approach allows decoding LoRa-like signals regardless of the CFO. Also, in terms of Dopplerrate, we basically tested the worst-case scenario in the context of LEO communication, since we considered a value of 280 Hz/s.

Reviewer 2 Report
The authors propose some enhancements to the physical layer of LORA-like transceivers in order to improve their tolerance to synchronization and frequency-shift errors, thus increasing their feasibility for LEO communication scenarios. The paper is reasonably well written and the proposal could be interesting. The main drawback is that the analysis is completely theoretical and the results based on MATLAB numerical simulations. An SDR implementarion and some real experiments in a controlled testbed could be interesting to definitely prove the validity of the author's proposal, but they are mentioned in the paper as a future work.
The paper focuses mainly on physical layer (modulation), but how other layers (MAC, for example) should be addapted to (or affected by) the LEO scenario is not discussed. I understand that is not the main topic of the paper, but maybe it deserves at least a couple or paragraph just to mention the issue and how it could be coped with (even if theoretically or by other authors).
In general the English of the paper is OK, but there are some "weird" constructions such as "breaking the glass". I suggest some english editing before the final publication of the manuscript.
Author Response
R: The authors propose some enhancements to the physical layer of LORA-like transceivers in order to improve their tolerance to synchronization and frequency-shift errors, thus increasing their feasibility for LEO communication scenarios. The paper is reasonably well written and the proposal could be interesting. The main drawback is that the analysis is completely theoretical and the results based on MATLAB numerical simulations. An SDR implementarion and some real experiments in a controlled testbed could be interesting to definitely prove the validity of the author's proposal, but they are mentioned in the paper as a future work.
A: We thank the reviewer for his comment. He has right to point out the importance of real-time communication via SDRs to validate the simulation results of our proposal. In fact, we are currently implementing this algorithm with C++ on SDR USRP B100. The goal of this processing is to perform real-time communication with a LEO satellite in a collaboration with Eutelsat. It should be noted that we are used to implement our algorithms on SDRs. Thus, we mentioned naturally SDR implementation as future works.
R: The paper focuses mainly on physical layer (modulation), but how other layers (MAC, for example) should be addapted to (or affected by) the LEO scenario is not discussed. I understand that is not the main topic of the paper, but maybe it deserves at least a couple or paragraph just to mention the issue and how it could be coped with (even if theoretically or by other authors).
A: We thank the reviewer for his suggestion. We provided some details on the adaptability of the LoRa MAC layer in LEO satellite scenario. Thus, the following paragraph was added in the introduction:
"Given this massive connectivity, the use of ALOHA-based random access protocols, as is the case of ground-based LoRa networks, is a challenging task. Indeed, in the IoT communication with LEO satellites, the satellite needs to serve many terminals at the same time, thus the probability of packet collision is very important in such scenario. Hence, an optimization of ALOHA could be implemented to reduce this conflict. For instance, authors in [5] propose to combine ALOHA with Time Division Multiple Access (TDMA) in order to reduce the probability of packet collision. However, this approach needs further synchronizations process between the satellite and end-devices, which would increase their energy consumption.
In the same context, in order to assure the latter synchronization process, only Class B and C of LoRaWAN [6] can be employed. Given that Class C mode increases considerably the energy consumption of end-devices, which only turns off the receiving window when it is sending data, authors in [5] propose to use Class B mode in this scenario. In this paper, we will deal only with the impact of Doppler effects when we consider LoRa-like communications with LEO satellites."
R: In general the English of the paper is OK, but there are some "weird" constructions such as "breaking the glass". I suggest some english editing before the final publication of the manuscript.
A: We thank the reviewer for his suggestion. We will change the construction "breaking the glass" by "bypass". Moreover, we performed a full check of the English structures in the manuscript.

Round 2
Reviewer 1 Report
Paper is ready for publication
Author Response
Dear reviewer,
Thanks for your effort to review the revised manuscript.